# Atmosphere-Dependent Electron Relaxation of the Ag-Decorated TiO$_2$ and the Relations with Photocatalytic Properties

Wenhao Zhao [1], Liping Wen [2] and Baoshun Liu [1,*]

1   State Key Laboratory of Silicate Materials for Architectures, Wuhan University of Technology, Wuhan 430070, China; wenhao1217@yeah.net
2   School of Environmental & Biological Engineering, Wuhan Technology and Business University, Wuhan 430065, China; awende@163.com
*   Correspondence: bshliu@whut.edu.cn

**Abstract:** In the current research, the atmosphere effects on the photoinduced electron relaxations of the undecorated TiO$_2$ and Ag-decorated TiO$_2$ (Ag/TiO$_2$) were carefully studied by means of the in situ photoconductance and diffuse reflection measurements. In pure N$_2$ atmosphere, the results showed that the electron relaxation mainly occurs through the transfer to the residual O$_2$, and the Ag nanoparticles form a fast electron transfer pathway. It was seen that the apparent activation energy of the electron transfer to O$_2$ was greatly reduced by the Ag decoration. In the methanol-containing N$_2$ atmosphere, the electron relaxation can still occur via the transfer to residual O$_2$ in the case of the undecorated TiO$_2$, while the relaxation mechanism changes for the Ag/TiO$_2$ as the relaxations are decreased with the temperatures. It is possible that the methanol molecule adsorbed on the Ag/TiO$_2$ perimeters could act as the bridge for the recombination of the holes and the electrons stored in the Ag nanoparticles. Reducing the Ag nanoparticle size from 15 nm to 3 nm can greatly increase the electron relaxations due to the increase in Ag dispersion and Ag/TiO$_2$ interconnection. Although the electron transfer to O$_2$ was increased, both the photocatalytic oxidations of acetone and isopropanol showed a decrease after the Ag decoration. The results indicated that the photocatalytic oxidation was not limited by the electron transfer to O$_2$. The increased electron transfer to O$_2$ contributed to the recombination around the Ag/TiO$_2$ perimeters, and the photocatalytic activities were decreased.

**Keywords:** Ag/TiO$_2$; diffusion reflection; photoconductance; electron relaxation; electron transfer; photocatalytic organic oxidations





## 1. Introduction

Heterogeneous photocatalysis has been an attractive solution that uses solar energy to drive chemical reactions, such as hydrogen production [1,2], CO$_2$ reduction [3,4], pollutant removal [5,6], and chemical reforming [7,8]. In principle, photocatalysis results from the photogenerated electron relaxation through the transfer to reactants such as O$_2$ and organics. Under gaseous reaction conditions, the holes are captured by organic molecules and the electrons mainly transfer to a good electron acceptor, such as an O$_2$ molecule. In the case of TiO$_2$ materials, it had been found that the hole transfer to organics occurs on the timescale of ps to ns, while the electron transfer to O$_2$ occurs on the timescale of ms to µs [9]. It is thus thought that the photocatalytic activity can be improved by increasing the electron transfer to O$_2$ through auxiliary catalysts and dopants.

Modification of TiO$_2$ materials with auxiliary catalysts is a general method to improve photocatalytic activity by changing charge carrier dynamics [10]. Auxiliaries include oxides, sulfides, carbides, and metals [11]. For example, Liu [12] et al. prepared a series of CeSmO$_x$-modified TiO$_2$ catalysts by the coprecipitation method and the impregnation method. The effect of the addition of Sm on the selective catalytic reduction of NH$_3$ (NH$_3$-SCR) was studied. It was found that the catalyst with the best molar ratio (Ce:Sm = 20:1) showed the

best deoxidation performance. Wang [13] et al. designed a novel glutathione photosensor based on $Bi_2S_3$-modified $TiO_2$ nanotube arrays; it was proved that the composite structure could accelerate the charge transfer. The sensitivity of the sensor to glutathione varies linearly from 15.0 to 200 $\mu mol \, L^{-1}$ under the excitation of visible light and 0.65 V bias voltage, showing good stability and repeatability. Fu [14] et al. synthesized graphene-modified $TiO_2$ aerogel composites by the hydrothermal method. It is confirmed that the synergistic effect between graphene bilayer capacitor and $TiO_2$ pseudo-capacitor, as well as the introduction of heteroatomic doping and defects, is conducive to improving the electrochemical performance of the material.

Metal nanoparticles, such as Ag [15], Au [16], Pt [17], Pd [18], and Ru [19], have been studied as the cocatalysts in $TiO_2$ photocatalysis. Transient studies [20] have shown that the photoinduced electrons in semiconductors could be directed to metal nanoparticles in a very short time; this can lead to effective charge carrier separation, prolong the lifetime of photogenerated carriers, enhance the activity of reactive species, and thus improve photocatalytic activities. For example, Dai [21] et al. monitored the decay kinetics of photoelectrons in P25 $TiO_2$ and $Au/TiO_2$ composites under UV excitation at 355 nm using nanosecond time-resolved infrared spectroscopy; the results showed that the transient infrared absorption of $TiO_2$ attenuates within 100 $\mu s$, while the corresponding time of $Au/TiO_2$ is only 1 $\mu s$. Akira [22] et al. observed the electron photogeneration in $TiO_2$ and $Pt/TiO_2$ catalysts using time-resolved infrared absorption spectroscopy; under the irradiation of 355 nm pump pulse, the transient infrared absorption energy of $Pt/TiO_2$ decays at subtle time scales. When $Pt/TiO_2$ is in contact with water, the light-generated holes oxidize the water within 2 $\mu s$, while the electrons reduce the water within 10~900 $\mu s$.

Interestingly, under UV irradiation, it had been seen that the photocatalytic activity of $Ag/TiO_2$ and $NiAg/TiO_2$ nanocore shells synthesized by Chuang [23] et al. was lower than that of $TiO_2$ nanoparticles, which might be ascribed to the low $TiO_2$ content and the electron transfer from $TiO_2$ shells to Ag or NiAg nuclei. At the same time, they can act as the recombination centers of photoinduced electrons and holes, resulting in the decrease in photocatalytic activity. We [24] had studied the role of Au as a co-catalyst in acetone photocatalysis, and found that the nano-Au modified $TiO_2$ did increase the electron transfer from $TiO_2$ to $O_2$, but the photocatalytic rate was decreased. Au modification also had no effect on the apparent activation energy of acetone photocatalytic oxidations. It was suggested that the Au-induced increase in electron transfer promotes recombination.

It had been found that Ag nanoparticles on the $TiO_2$ surface can serve as the sites for rapid electron capturing [25], but whether the electron transfer to Ag can increase the subsequent relaxation kinetics and catalytic activities was not well known. Furthermore, most studies focused on liquid phase photocatalytic reactions, and few of them devoted to explaining the Ag effect on photoinduced electron relaxation and their connection to photocatalytic activities under gas phase conditions. In the case of organic oxidations, the electron transfer to $O_2$ is a necessary step. Although it had been generally thought that increasing the electron transfer to $O_2$ should favor the photocatalytic activities, few works studied the direct relation between them. In this study, combined with in situ optical and electrical measurements, the role of Ag nanoparticle catalysts in the electronic relaxation of $TiO_2$ through the transfer to $O_2$ was investigated under different atmospheres, and whether the relaxation change could contribute to the photocatalytic pathway was discussed. It had also been [26] reported that the quantum efficiency of Ag-decorated $TiO_2$ ($Ag/TiO_2$) materials depended on the Ag sizes; the results showed that decreasing Ag particle size could increase the efficiency. Therefore, the effect of Ag nanoparticle size on the electron relaxations and photocatalytic activities was also discussed. The issue investigated in the current research gave a deep insight on the photocatalytic oxidation mechanism of the $Ag/TiO_2$. The experimental method and scientific finding of the current research should also be suitable for other co-catalyst modified photocatalyst, which should be meaningful for the design of highly-active photocatalysts.

## 2. Results and Discussion

### 2.1. Physical Characterizations

Figure 1A shows the TEM images of L-Ag/TiO$_2$ and EDX mappings of Ag, Ti, and O elements. It can be seen that some spherical nanoparticles are distributed among the TiO$_2$ nanoparticles (the red circle), and the high-resolution TEM image (top right corner) shows (111) lattice fringes of an Ag metal nanoparticle, which is also confirmed by the EDX element mapping. The size of the Ag nanoparticles in L-Ag/TiO$_2$ is about 15 nm. Figure 1B shows the TEM image and EDX mapping of S-Ag/TiO$_2$, and it can be seen that some small dots are distributed over the TiO$_2$ nanoparticles, and the (111) lattice fringes of the Ag metal nanoparticles are also shown in the upper right high-resolution TEM image. The EDX mapping also confirmed that these dots are small Ag nanoparticles. The size of the small Ag nanoparticles is about 3 nm. In addition, the high-resolution TEM image shown in the right-top corners of Figure 1A,B also clearly show the (101) and (110) characteristic crystal plane fringes of anatase and rutile TiO$_2$, respectively. It can also be seen that the Ag nanoparticles tightly connect with TiO$_2$ nanoparticles, which should favor the rapid electron transfer between them. The XRD patterns (Figure S2) show that the L-Ag/TiO$_2$ does not have the characteristic XRD peaks of metallic Ag and Ag oxides due to the low loaded capacity. According to the ICP-OES result, the corresponding contents of Ag in L-Ag/TiO$_2$ and S-Ag/TiO$_2$ was about 0.34 and 0.36 wt. %, respectively. Figure 2 shows the UV–Vis–NIR absorption spectra of the undecorated TiO$_2$, L-Ag/TiO$_2$, and S-Ag/TiO$_2$. The absorption threshold of the TiO$_2$ is about 410 nm; this corresponds to the interband transition absorption of the rutile TiO$_2$ phase in P25. Ag's deposition on the TiO$_2$ surface does not result in a clear change in the absorption threshold. The Ag/TiO$_2$ exhibits strong local surface plasmon resonance extinction, showing the metallic feature. The plasmon peak of the S-Ag/TiO$_2$ is wider and slightly asymmetrical than that of the L-Ag/TiO$_2$.

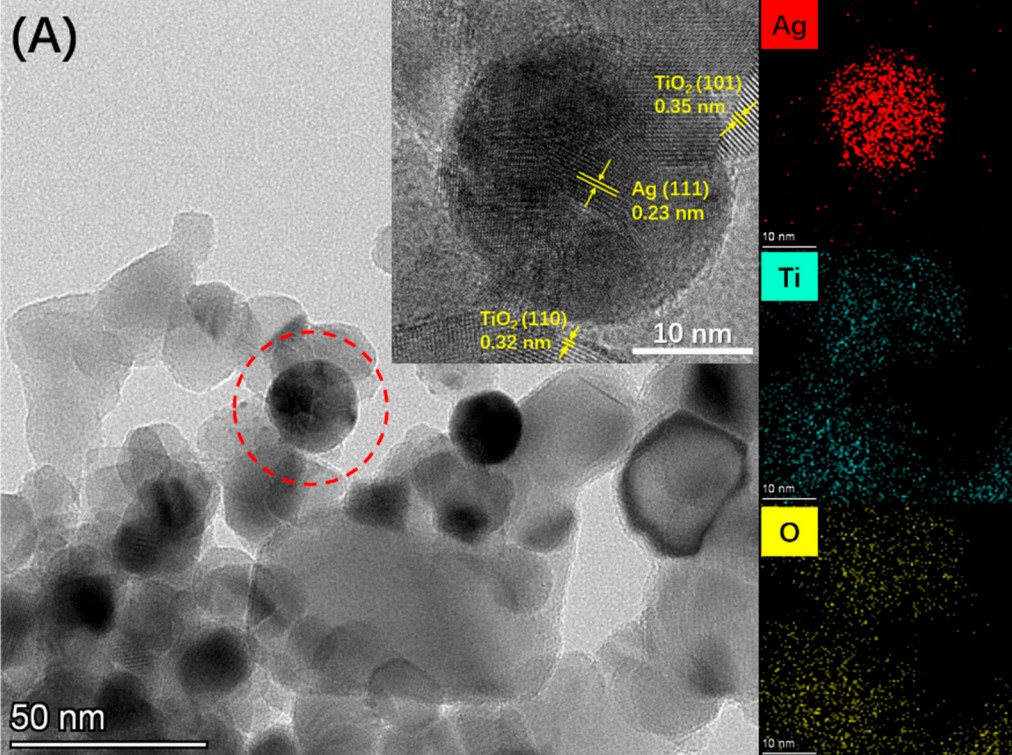

**Figure 1.** *Cont.*

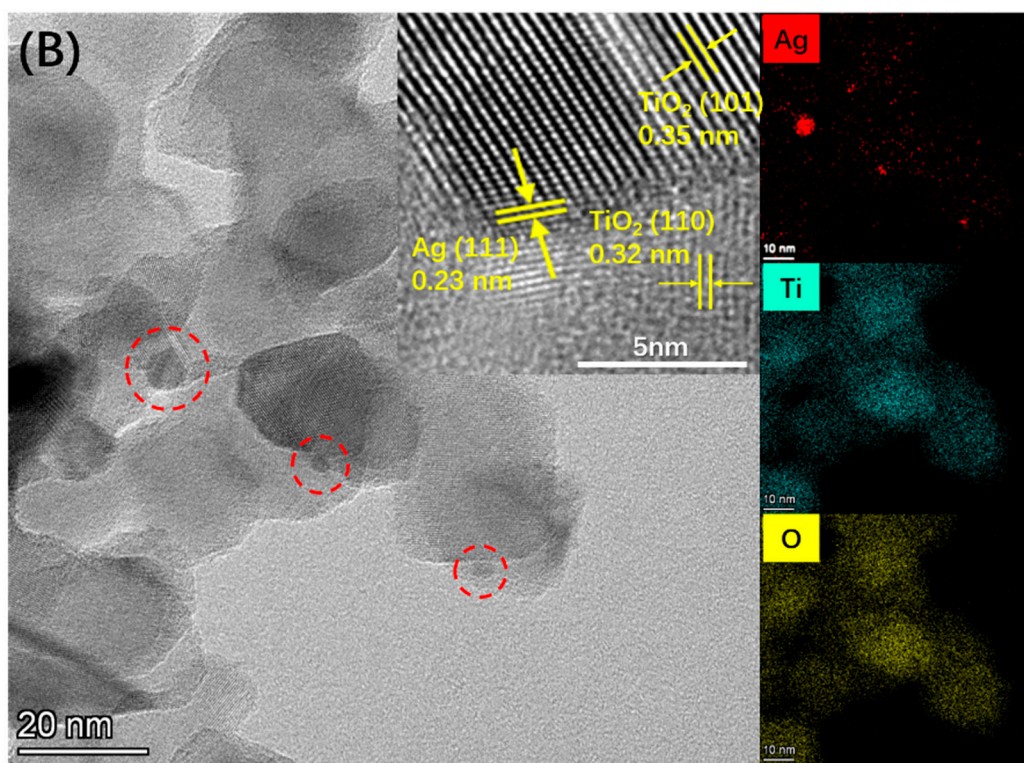

**Figure 1.** HR-TEM images and EDX mappings of (**A**) the L-Ag/TiO$_2$ and (**B**) the S-Ag/TiO$_2$ (red circles label the Ag nanoparticles).

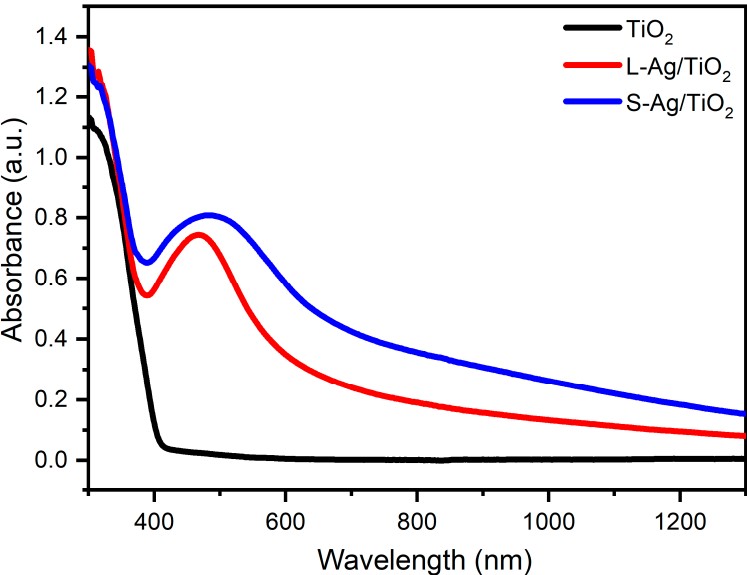

**Figure 2.** UV–Vis–NIR absorption spectra of the undecorated TiO$_2$, L-Ag/TiO$_2$, and S-Ag/TiO$_2$.

### 2.2. Atmosphere-Dependent Photoconductances and Relaxation Kinetics

2.2.1. Photoconductances in Pure N$_2$ Atmosphere

Since the loaded Ag amount of is very low, and they have good dispersion on the TiO$_2$ surfaces, the TiO$_2$ network is the medium for electron transport. All samples in this study show ohmic contact (Figure S3). It had also been revealed that the electron mobility is unchanged under light illumination for nano-TiO$_2$ materials. Therefore, the conductance change can be proportional to the electron density change, and the conductance relaxation can also reflect the relaxation of the photoinduced electrons. In pure N$_2$ atmosphere, the photocurrent variations of the undecorated TiO$_2$ with time (Figure S4A) show that the

steady-state photocurrents decrease with the measured temperatures. The photocurrents of L-Ag/TiO$_2$ and S-Ag/TiO$_2$ over time (Figure S4B,C) show that their rising behaviors are different, and the steady-state photocurrents also decrease with the increase in temperature. In addition, the steady-state photocurrents of the Ag/TiO$_2$ are lower than that of the unmodified TiO$_2$. Especially for S-Ag/TiO$_2$, the steady-state photocurrents can be seen to be reduced by tens of times.

Figure 3 shows the normalized photocurrent relaxations of the undecorated TiO$_2$ (A), L-Ag/TiO$_2$ (B), S-Ag/TiO$_2$ (C) in pure N$_2$ atmosphere, respectively. It can be seen that all the relaxations increase with the temperatures, and the increase becomes slower for the Ag/TiO$_2$ samples. The $k_{et}$ values of the photocurrent relaxations just after the light illumination were obtained based on the first-order kinetics, with the Arrhenius plots of the $k_{et}$ being shown in Figure 3D. It can be seen that the Ag decoration can increase the electron relaxations, and the smaller the Ag particle size, the faster the relaxations; this is ascribed to the higher loaded amount and high dispersion over TiO$_2$ surface. The $k_{et}$ of the three samples increases with the temperatures, based on which the activation energies ($E_{app}$s) of the relaxations are determined to be 24 kJ/mol, 14 kJ/mol, and 10 kJ/mol for the undecorated TiO$_2$, L-Ag/TiO$_2$, and S-Ag/TiO$_2$, respectively. It can be seen that the Ag decoration reduces the $E_{app}$s of the electron relaxations, indicating that the electron relaxation pathway is changed.

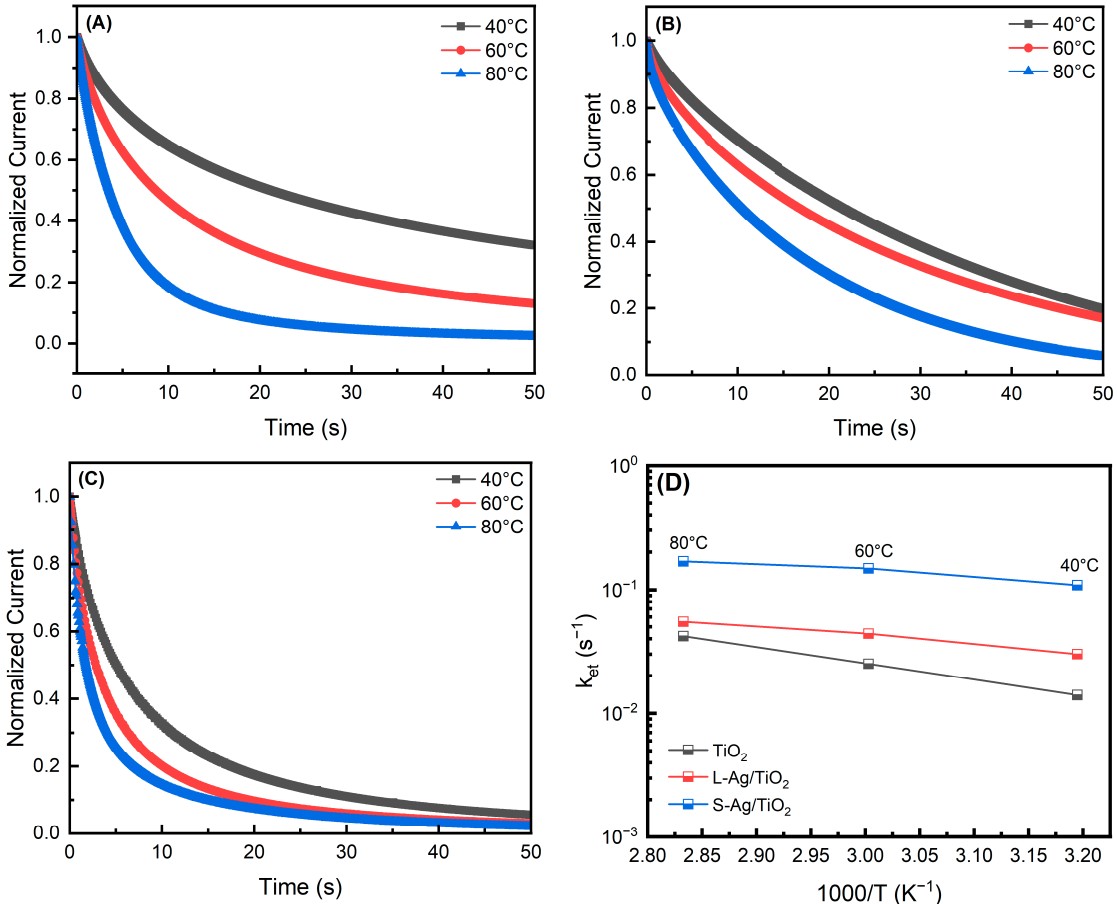

**Figure 3.** Normalized photocurrent relaxations of the undecorated TiO$_2$ (**A**), L-Ag/TiO$_2$ (**B**), and S-Ag/TiO$_2$ (**C**) in pure N$_2$ atmosphere. (**D**) Temperature dependences of the conductance relaxation kinetic constant $k_{et}$ in N$_2$ atmosphere.

In the case of the Ag/TiO$_2$, the electron relaxation should consist of two parallel pathways, namely, the relaxation involving Ag sites and without involving Ag sites. Therefore, the current relaxation is

$$\frac{dI(t)}{dt} = -k_{et,1}I(t) - k_{et,2}I(t) = -(k_{et,1} + k_{et,2})I(t),\tag{1}$$

where $k_{et,1}$ and $k_{et,2}$ are rate constants of the current relaxation involving and without involving the Ag sites, respectively. Thus, the contribution of the electron relaxations via the Ag sites under steady state can be estimated by

$$\Gamma_{Ag} = 1 - \frac{k_{et,2}}{k_{et,1} + k_{et,2}}.\tag{2}$$

Based on the data shown in Figure 3D, it was discovered that ~53% (95%), 43% (83%), and 23% (75%) of the electrons in the L-Ag/TiO$_2$ (S-Ag/TiO$_2$) relaxed through the Ag-assisted pathway at 40, 60, and 80 °C, respectively.

### 2.2.2. Photoconductances in the Methanol-Containing N$_2$ Atmosphere

Time-dependent photocurrents of the undecorated TiO$_2$, L-Ag/TiO$_2$, and S-Ag/TiO$_2$ in methanol-containing N$_2$ atmosphere were also obtained (Figure S5A–C). The results show that the photocurrent rising slopes become faster after the Ag decoration. In addition, the steady-state photocurrent is much higher than that in pure N$_2$. The results show that the upslope of photocurrent increases after silver plating. In addition, the steady-state photocurrent is much higher than that of pure N$_2$ due to the rapid hole trapping by methanol molecules. It can also be seen that the steady-state photocurrent decreases with the temperatures for the undecorated TiO$_2$ but increases for the Ag/TiO$_2$ samples. Figure 4A–C show the normalized photocurrent relaxations of the three samples, respectively. The $k_{et}$ values of the photocurrent relaxations just after the light illumination were also obtained, with Arrhenius plots being shown in Figure 4D. It can be seen that the $k_{et}$ values of the undecorated TiO$_2$ are similar to that in N$_2$ atmosphere. The $E_{app}$ of the electron relaxation is ~21 kJ/mol, this is close to that in pure N$_2$, showing that the methanol presence would not affect the electron relaxation mechanism. The photocurrent relaxations of the Ag/TiO$_2$ become much faster than that in the N$_2$ atmosphere after illumination termination. The $k_{et}$ values of the Ag/TiO$_2$ are almost two orders of magnitude higher than that of the undecorated TiO$_2$ at room temperature. It is also interesting to see that the relaxation $k_{et}$ of the Ag/TiO$_2$ decreases with the temperatures, meaning that the Ag-assisted electron relaxation pathway is thermally prohibited in the presence of methanol and is different from that in pure N$_2$. According to Equation (2), the percentages of the electron relaxation assisted by the Ag sites are calculated, which are 99.3% (99.4%), 96.7% (97.7%), 87.7% (92.5%), and 80% (84.7%) in the Ag/TiO$_2$ (S-Ag/TiO$_2$) at 20, 40, 60, and 80 °C, respectively. It can be seen that almost 100% of the electrons can relax through the Ag sites at low temperatures.

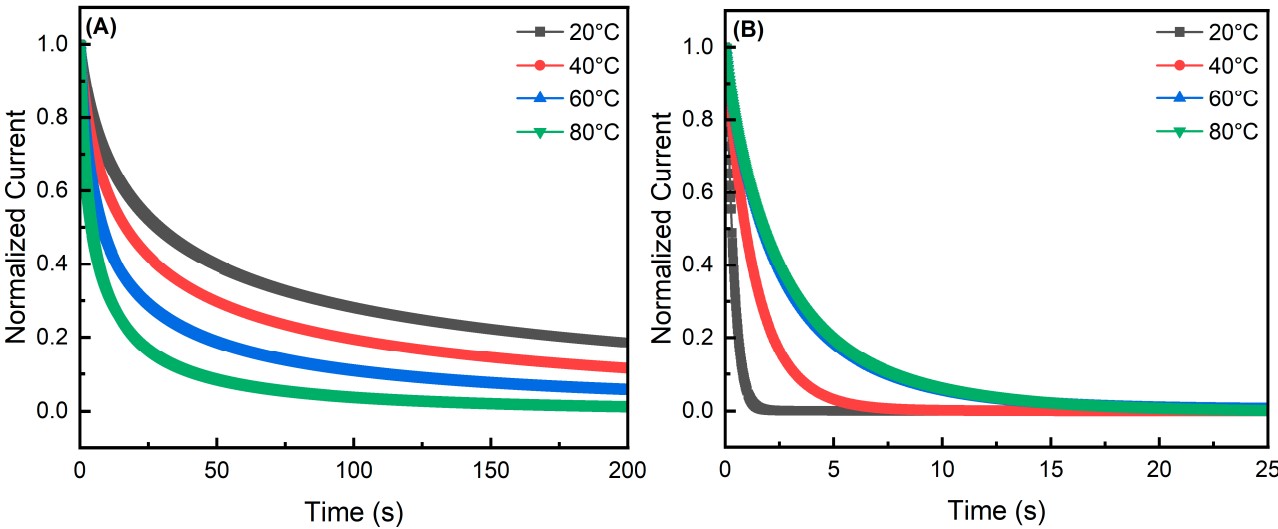

**Figure 4.** *Cont.*

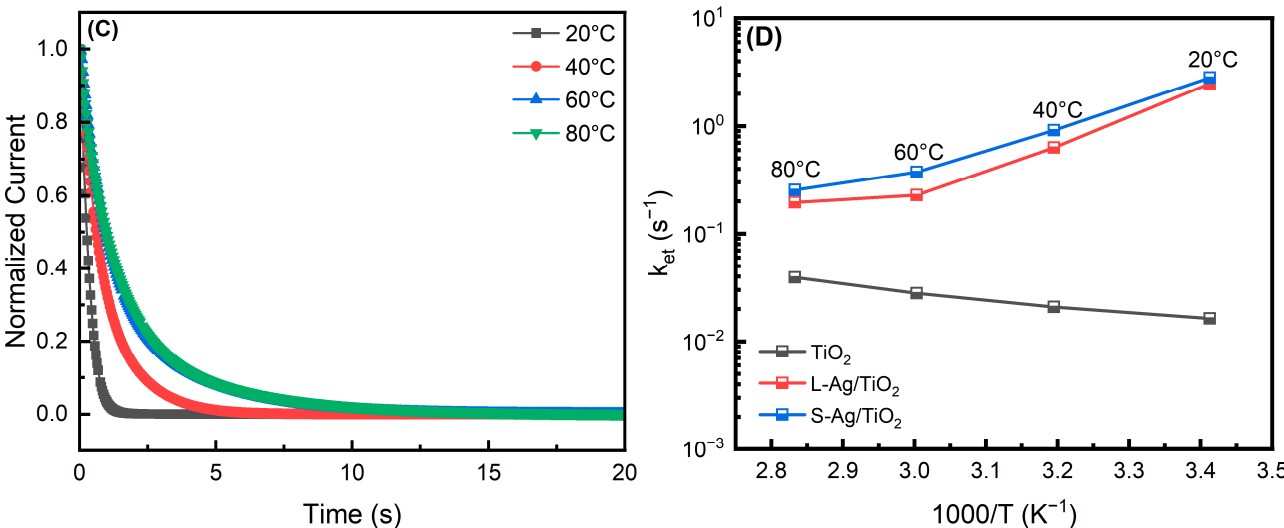

**Figure 4.** Normalized photocurrent relaxations of the undecorated TiO$_2$ (**A**), L-Ag/TiO$_2$ (**B**), S-Ag/TiO$_2$ (**C**) in the methanol-containing N$_2$ atmosphere. (**D**) Temperature dependences of the photocurrent relaxation $k_{et}$ for the three samples.

*2.3. Atmosphere-Dependent Photoinduced Absorption and Relaxation Kinetics*

2.3.1. Diffusion Absorption in N$_2$ Atmosphere

It had been shown that the photoinduced absorption can reflect the electron accumulation in TiO$_2$ [27]. Thus, the absorption of the undecorated TiO$_2$, L-Ag/TiO$_2$, and S-Ag/TiO$_2$ at 1550 nm under and after the UV light illumination were also checked at different temperatures in pure N$_2$ atmosphere to show the effect of Ag decoration on electron relaxation. The time-dependent absorptions of the undecorated sample at different temperatures (Figure S6A) shows that the steady-state absorptions decrease with the temperatures. The absorptions relax after the end of the light illumination, with its rate increasing with the temperatures; this result is in good consistency with the photocurrent analysis. The time-dependent absorptions of the L-Ag/TiO$_2$ at different temperatures (Figure S6B) also show that the steady-state absorptions decrease with the temperatures, indicating that increasing temperatures can increase the electron relaxation after Ag decoration. Figure 5A compares the photoinduced absorption of the undecorated TiO$_2$, L-Ag/TiO$_2$, and S-Ag/TiO$_2$ at ~20 °C, it can be seen that the steady-state absorptions are greatly decreased by Ag decoration, meaning the great increase in the electron relaxation. Actually, after the end of light illumination, it can be seen that the electron relaxation of the L-Ag/TiO$_2$ is higher than that of pure TiO$_2$. The photoinduced absorption of the S-Ag/TiO$_2$ could not be even seen; this indicates that the electrons cannot accumulate due to the very fast electron relaxation in this sample.

Figure 5B,C shows the normalized photoabsorption relaxations at different temperatures for the undecorated TiO$_2$ and L-Ag/TiO$_2$. The $k_{et}$ values of the absorption relaxations just after the end of the light illumination are obtained from the quasi-first order kinetics based on the normalized absorptions. Figure 5D shows the Arrhenius plots of the absorption relaxation $k_{et}$ of the undecorated TiO$_2$ and L-Ag/TiO$_2$ in pure N$_2$ atmosphere, and the $k_{et}$ of the S-Ag/TiO$_2$ cannot be obtained. The $E_{app}$s of the electron relaxations for the undecorated TiO$_2$ and L-Ag/TiO$_2$ are determined to be ~21 kJ/mol and 8 kJ/mol, respectively; this indicates the change of the electron relaxation pathway by the Ag decoration. Similarly, based on the Equations (1) and (2), the contribution of the electron relaxations via the Ag-assisted pathway are ~78%, 75%, 66%, and 45% for the L-Ag/TiO$_2$ at 20, 40, 60, and 80 °C, respectively. The results obtained from the photoinduced relaxation agree well with the that obtained from the photoconductance.

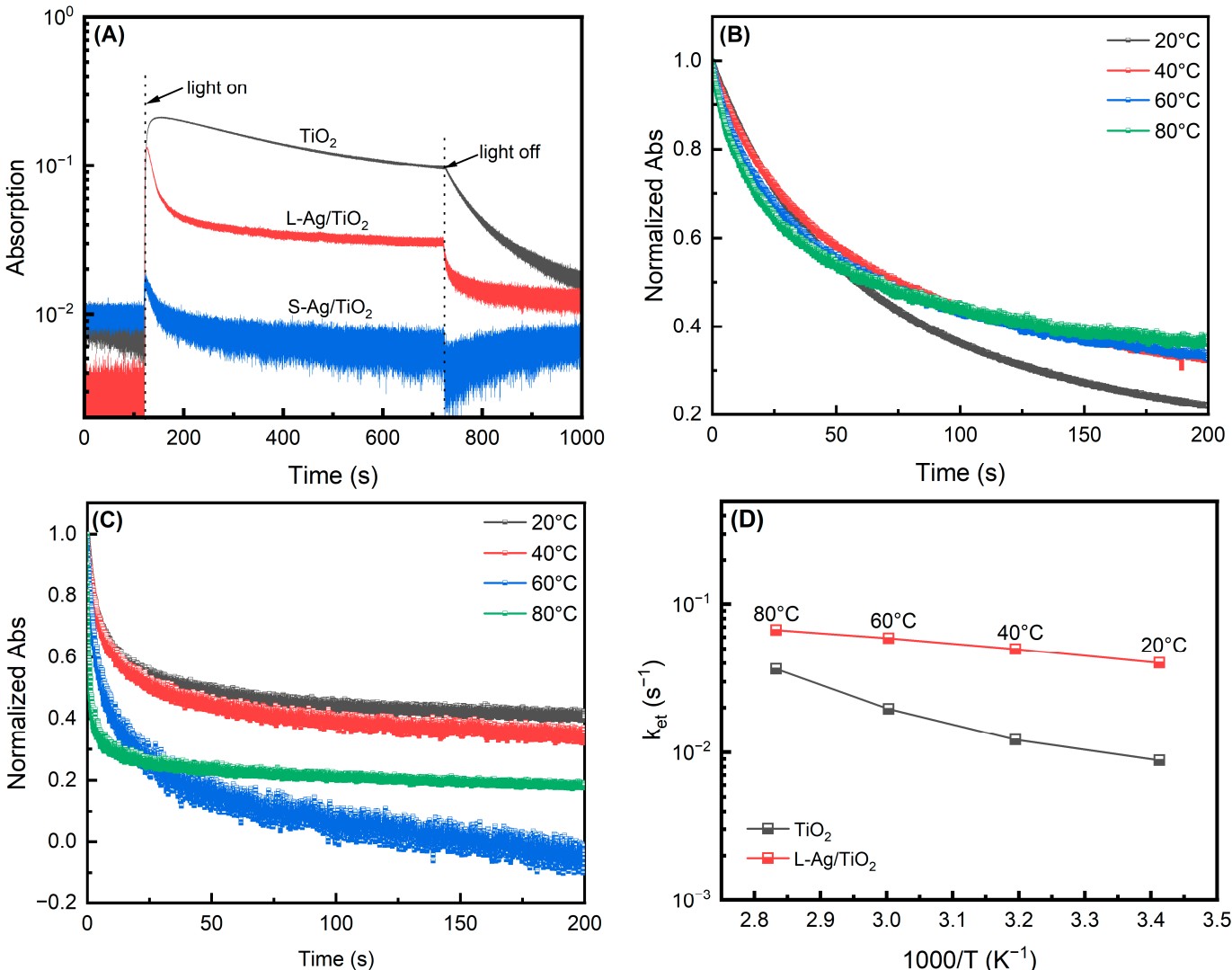

**Figure 5.** (**A**) Absorption changes of the three samples at 1550 nm before and after 375 nm laser irradiation in $N_2$ atmosphere at room temperature; normalized optical absorption relaxation of the undecorated $TiO_2$ (**B**) and L-Ag/$TiO_2$ (**C**) in $N_2$ atmosphere. (**D**) Temperature dependences of the optical absorption relaxation kinetic constant $k_{et}$ in $N_2$ atmosphere for the undecorated $TiO_2$ and L-Ag/$TiO_2$.

### 2.3.2. Photoinduced Absorption in the Methanol-Containing $N_2$ Atmosphere

The time-dependent absorptions of the undecorated $TiO_2$, L-Ag/$TiO_2$, and S-Ag/$TiO_2$ at 1550 nm were also obtained at different temperatures in the methanol-containing $N_2$ atmosphere (Figure S7A–C). The results show that the steady-state absorptions of the undecorated $TiO_2$ decreases, while that of the L-Ag/$TiO_2$ and S-Ag/$TiO_2$ increases with the temperature. In addition, the steady-state absorption decreases after the Ag decorations. Especially for S-Ag/$TiO_2$, the photoinduced absorption was reduced by tens of times, which is consistent with the above photocurrent analysis. It can be seen that the absorption relaxations are greatly increased by the Ag nanoparticles when methanol is present. Figure 6A–C shows the normalized photoinduced absorptions just after the light illumination; this shows that the relaxation increases with the temperatures for the undecorated $TiO_2$ but decreases for the Ag-decorated $TiO_2$.

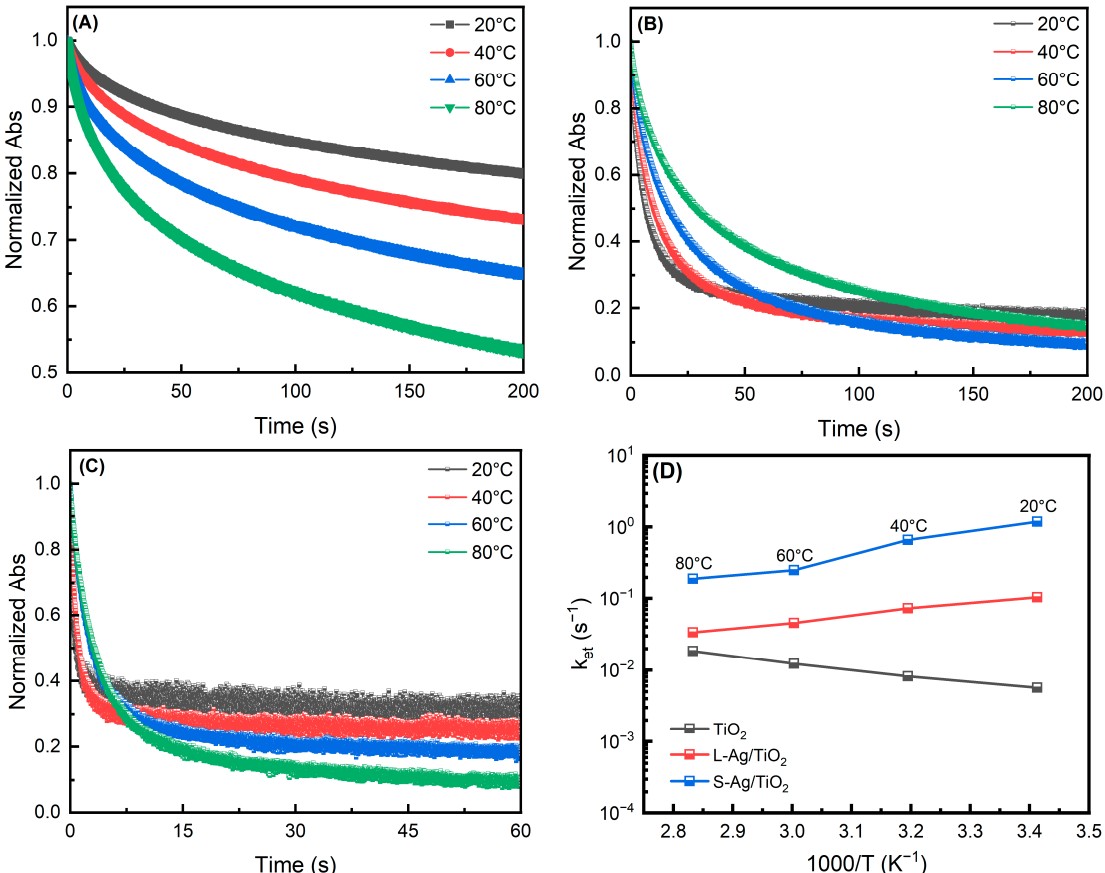

**Figure 6.** Normalized optical absorption relaxations of the undecorated $TiO_2$ (**A**), L-Ag/$TiO_2$ (**B**), and S-Ag/$TiO_2$ (**C**) in $N_2$ atmosphere containing methanol. (**D**) Temperature dependences of the optical absorption relaxation kinetic constant $k_{et}$ in methanol atmosphere.

　　　Similarly, the relaxation $k_{et}$ values are obtained from the first-order kinetics based on these data, and the Arrhenius plots are shown in Figure 6D. It can be seen that the Ag decoration increases the electron relaxations at all the studied temperatures, and the role of small Ag nanoparticles is much higher. At ambient temperatures, the $k_{et}$ is increased by more than 10 times and 100 times after decoration with the large and small Ag nanoparticles. The absorption relaxations are also characterized by thermal activation for the undecorated $TiO_2$, and the $E_{app}$ of the relaxation is ~20 kJ/mol; this is almost the same as that obtained in pure $N_2$. Therefore, it can be known that the methanol exposure should not change the electron relaxation mechanism for the undecorated sample, consistent with the photocurrent tests and previous studies [24]. However, the $k_{et}$ values of the Ag-$TiO_2$ becomes decreased with the temperatures; this means that the kinetics of the electron relaxations becomes negatively dependent on temperature when methanol is added; this phenomenon is in accordance with the above conductance analysis and our previous study [28].

　　　Based on the Equations (1) and (2), the contributions to the electron relaxation via the Ag-assisted pathway in the methanol-containing $N_2$ atmosphere are also estimated: ~95% (100%), 85.3% (99%), 72% (95%), and 45% (95%) of the total electrons can relax via the Ag sites of the L-Ag/$TiO_2$ (S-Ag/$TiO_2$). In particular, the electron relaxation of the S-Ag/$TiO_2$ mainly transfers via the Ag sites at all temperatures, also agreeing well the conductance analysis. In addition, given that the electron relaxation in general decreases with electron densities, the actual contribution of Ag-assisted relaxation should be higher because of the much lower electron density in Ag-modified $TiO_2$, as compared to the undecorated $TiO_2$. Thus, it is reasonable to assume that almost 100% of the total electrons could relax via the Ag-assisted pathway for S-Ag/$TiO_2$.

### 2.4. Relaxation Pathways

The above photoconductance and absorption analysis showed that the electron relaxations are increased by Ag nanoparticles in pure $N_2$ and methanol-containing $N_2$ atmosphere. Then, what pathway did they occur? The photocurrents were also measured under vacuum at different temperatures (Figure S8A–C) to reveal the mechanism. Figure 7A compares the photocurrent relaxation of the three samples at 80 °C under vacuum. It was seen that the electron relaxations cannot be greatly increased after Ag decoration; this is different from that obtained in pure $N_2$ and methanol-containing $N_2$ atmospheres. Based on the normalization of the relaxations, the $k_{et}$ values were obtained according to the same manner, and the Arrhenius plots are shown in Figure 7B; this shows that the $k_{et}$ increases with the temperatures. The $E_{app}$s for the current relaxations of the undecorated $TiO_2$, L-Ag/$TiO_2$, and S-Ag/$TiO_2$ are ~14 kJ/mol, ~10 kJ/mol, and ~12 kJ/mol, respectively; this shows that there is not a difference among them. It had been revealed that the electron relaxation under vacuum mainly occurs through the transfer to residual $O_2$ [28]. Due to very low residual $O_2$ (about $10^{-3}$ Pa), the $O_2$ transport to the Ag sites is greatly limited, so the contribution of the Ag sites to the electron relaxation cannot be high. Therefore, the electrons in the Ag–$TiO_2$ mainly relax through the surfaces of $TiO_2$ in this case. However, the photocurrent relaxations of the Ag/$TiO_2$ in pure $N_2$ are much higher than that of pure $TiO_2$; this can be ascribed to the high residual $O_2$ in $N_2$ atmosphere that can greatly increase the $O_2$ transport to Ag sites. It had been seen that the electron transfer to $O_2$ is thermally activated, so the $k_{et}$ of the electron transfer from the Ag/$TiO_2$ to $O_2$ also increases with the temperatures. From this viewpoint, it can be known that the Ag sites provide an alternative fast pathway for the electron transfer to $O_2$. However, in the methanol-containing $N_2$ atmosphere, firstly, it was seen that the electron relaxation is faster; secondly, the relaxation decreases with the temperatures. Therefore, electron relaxation should not mainly occur via the transfer to $O_2$, and a new relaxation pathway might be formed. For example, it had been reported that the Au/$TiO_2$ perimeter can form hole accumulation regions [29], this should be similar to the Ag/$TiO_2$. In this case, the methanol molecule can capture the holes in the Ag/$TiO_2$ perimeter to form a positive methyl group nearby Ag sites. The electrons stored in the Ag nanoparticles can then directly transfer to the hole-captured methyl groups and go on to recombination; increasing the temperatures would decrease the recombination rate, possibly due to a change in methanol-Ag interaction. In our previous study [30], we have also seen that in situ formed metallic Cu sites can lead to a great increase in electron relaxations in the methanol-containing $N_2$, which also decrease with the temperatures; this can be attributed to the same recombination mechanism. A similar phenomenon was also observed by us for the metallic Au nanoparticle-modified $TiO_2$ in recent experiments. Therefore, this should be common for the metallic nanoparticle decorated $TiO_2$, and the detailed mechanism is still under investigation.

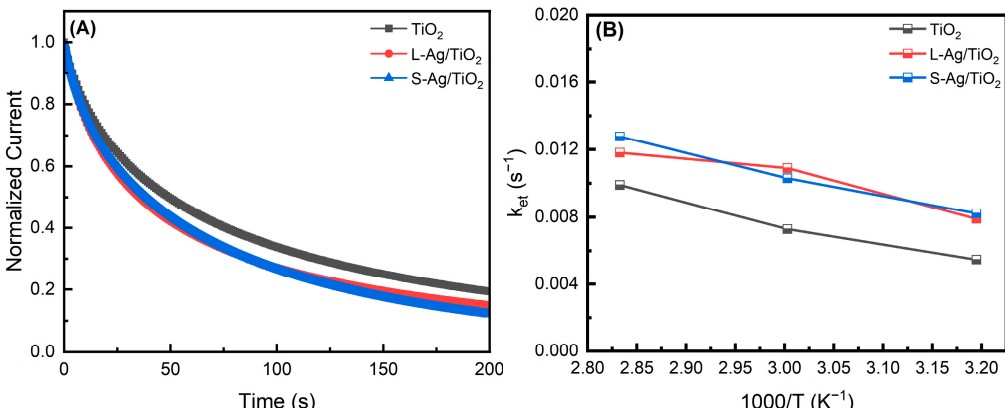

**Figure 7.** (**A**) Comparison of current relaxations of the undecorated $TiO_2$, L-Ag/$TiO_2$ and S-Ag/$TiO_2$ under vacuum at 80 °C. (**B**) Temperature dependencies of the kinetic constant $k_{et}$ of the current relaxations.

### 2.5. Photocatalytic Properties and the Relation with the Electron Relaxation

The $CO_2$ release curves during the acetone photocatalytic oxidations by the undecorated $TiO_2$, L-Ag/$TiO_2$, and S-Ag/$TiO_2$ under UV light illumination at different temperatures (Figure S9A–C) were used to evaluate the photocatalytic activities. It can be seen that the $CO_2$ generations almost follow the zero-order kinetics and increase with the reaction temperatures. The rate constants ($k_{CO_2}$) of the $CO_2$ generations are shown in Figure 8A in Arrhenius form for the different samples; this shows that the photocatalytic activity of L-Ag/$TiO_2$ and S-Ag/$TiO_2$ is lower than that of the undecorated $TiO_2$ at all reaction temperatures. Figure 8B shows the conversion of isopropanol to acetone over the undecorated $TiO_2$, L-Ag/$TiO_2$, and S-Ag/$TiO_2$ at different temperatures. The results also show that the catalytic activities of the L-Ag/$TiO_2$ and S-Ag/$TiO_2$ slightly decreases for the Ag/$TiO_2$ samples. It can be seen that the photocatalytic activities also increase with the temperatures, but the effects of the temperatures are lower as compared to the photocatalytic acetone oxidation.

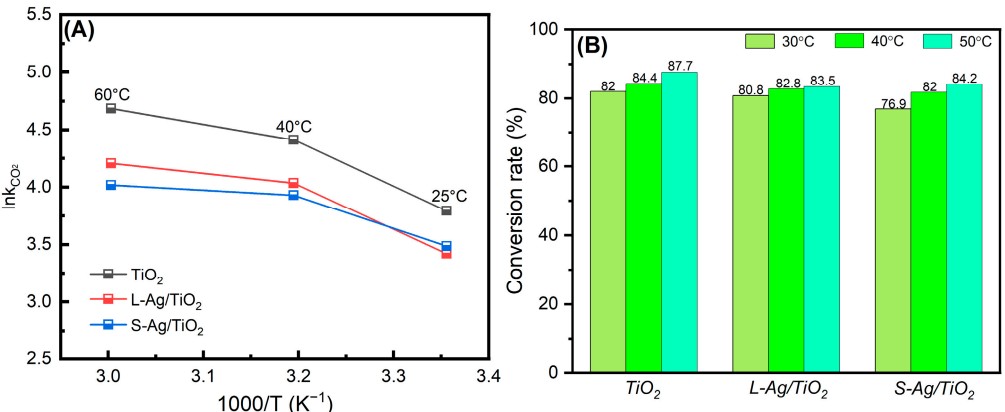

**Figure 8.** (**A**) Temperature dependences of the kinetic constants of the $CO_2$ generations for acetone oxidations. (**B**) Isopropanol conversions of the undecorated $TiO_2$, L-Ag/$TiO_2$, and S-Ag/$TiO_2$ at different temperatures.

The results show that the Ag decoration cannot increase both the acetone and isopropanol photocatalytic oxidations, although the electron transfer to $O_2$ was increased. As a good electron capturer, $O_2$ can first react with electrons to generate $O^{2-}$, while the holes are captured by acetone or isopropanol molecule. The reaction of $O^{2-}$ with the hole trapped organic molecule can lead to photocatalytic oxidations. It is thus expected that the increase in electron transfer via the Ag nanoparticles can improve the photocatalytic activities. However, the activity decrease means that the increased electron transfer to $O_2$ cannot contribute to the photocatalytic oxidations. Furthermore, the temperature effects on the acetone and isopropanol oxidation are almost the same for the three samples. From this viewpoint, the electron transfer to $O_2$ cannot be the limited step for photocatalysis, although it is relatively slow, which is also independent of Ag deposition. As revealed in our previous study [24,30], the electron transfer to $O_2$ can also contribute to the recombination in addition to the photocatalytic effect; it can be thus considered that Ag nanoparticles must increase the recombination. It had been reported [29] that the photoinduced holes are more prone to accumulated in the Au/$TiO_2$ perimeter. In this case, the electrons transferring to Ag nanoparticles might recombine with the holes around the Ag/$TiO_2$ perimeter through the $O_2$ adsorption–desorption cycle, as illustrated in our previous study [24]. Since the Ag decoration has no significant effect on the temperature dependence of the acetone and isopropanol photocatalysis, the holes near the Ag/$TiO_2$ interface do not, in the main, participate in photocatalysis but go on to recombination. The results also indicate the acetone and isopropanol molecule should mainly reside over the $TiO_2$ surfaces. The photocatalytic oxidation should mainly occur over the $TiO_2$ surfaces, and some $O^{2-}$ formed over the Ag sites should transport to the $TiO_2$ surfaces to complete the photocatalytic oxidation. The

above catalytic mechanism is shown in Figure 9; this is also consistence with our previous studies [24,28,30].

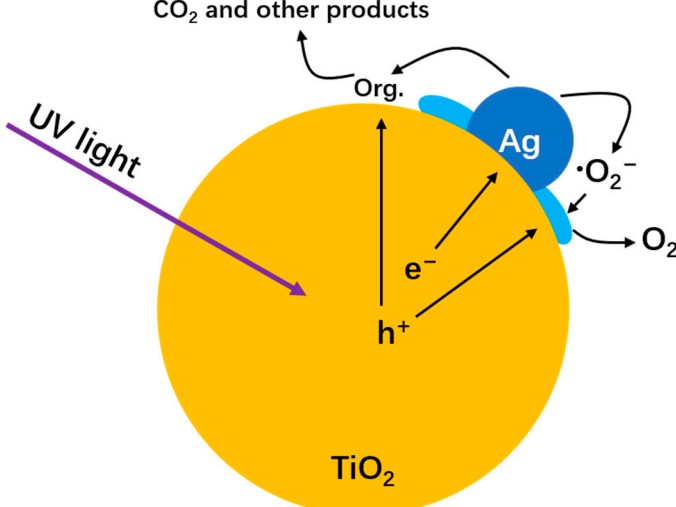

**Figure 9.** Electron relaxation pathways and photocatalytic oxidation Mechanism for the Ag-decorated $TiO_2$.

## 3. Experimental Section

### 3.1. Sample Preparation

Large Ag nanoparticle-modified $TiO_2$ (L-Ag/$TiO_2$) was prepared through photode-position [31] by adding 0.1 M $AgNO_3$ aqueous solution (92.7 μL) to methanol aqueous solution (50 mL, 20%, $v/v$) and then adding 0.2 g pre-grounded P25 $TiO_2$ powder. After continuously stirring in the dark for 40 min, the solution was irradiated for 2 h with a 500 W Xe lamp at a constant $N_2$ flow rate. After irradiation, it was cleaned and precipitated with distilled water and anhydrous ethanol several times. After adding ethanol, it was ultrasonically dispersed in a glass container. Finally, it was dried at 60 °C in $N_2$ atmosphere for 4 h. The preparation process of the small size Ag nanoparticle modified $TiO_2$ material (S-Ag/$TiO_2$) was similar to that of L-Ag/$TiO_2$. After $AgNO_3$ solution was added, equal molar amount of NaCl solution was added, and then 0.1 M of NaOH solution was used to adjust the pH of the reaction solution to ~11 [32]. The unmodified $TiO_2$ samples used in the control experiment were prepared following the same procedure as L-Ag/$TiO_2$, except that $AgNO_3$ was not used.

### 3.2. Catalyst Characterization

The morphologies of the sample and the combination of Ag and $TiO_2$ were studied by field emission transmission electron microscopy (TEM; Model: Talos F200S, Waltham, MA, USA). An X-ray diffractometer (XRD; Empyrean, PANalytical, Almelo, The Netherland) was used to study crystal structures in which Cu Kα radiation was used as an X-ray source. The percentage of Ag nanoparticles supported on $TiO_2$ was determined by inductively coupled plasma optical emission spectrometry (ICP-OES, Prodigy7, LEEMAN LABS, Hudson, NH, USA). The ultraviolet–visible diffuse reflection spectrum was measured by an ultraviolet–visible spectrophotometer equipped with an optical integrating sphere with a wavelength range from 300 nm to 1300 nm (UV-2600, Shimadzu, Kyoto, Tokyo, Japan).

### 3.3. In Situ Photoconductance Measurement

On the basis of our previous work [30], in situ photoconductances of pure $TiO_2$ and Ag/$TiO_2$ samples were obtained in a self-designed device, the schematic diagram of which is shown in Figure S1. First, the 0.05 mm FTO strip in the center of the 20 mm × 20 mm FTO glass was removed by laser etching, and the transparent tape was fixed on both sides of the etched strip to form a slit wider than it. Then pure $TiO_2$ and Ag/$TiO_2$ paste were applied on the slit

area, and the tape was removed after scraping with a scraper. It is then dried at 50 °C to form a sample coating. At various temperatures, a 2 V bias voltage was applied using a Keithley-2450 source meter, and conductances were measured in a dual probe mode using a DMM7510. In the case of the vacuum atmosphere photoconductance measurement, the chamber vacuum degree was pumped to 0.02 Pa by a vacuum molecular pump (PFEIFFER VACUUM-Air-TC110, Asslar, Germany). During atmospheric photoconductance measurements, pure $N_2$ or methanol-containing $N_2$ passed through the reaction chamber at a flow rate of 0.3 NL/min under the control of a flowmeter and continues to flow for approximately 30 min prior to the test to ensure that the chamber was filled with $N_2$ or methanol gas. The conductances of the coating were firstly measured in the dark for 5 min, then monitored for 10 min under monochromatic 375 nm laser irradiation, and finally continued to be measured for 5 min after the laser irradiation stopped to obtain the time-current curve. A silicon-based photodetector (Newport 843-R) is used to check the light intensity.

### 3.4. In Situ Diffuse Reflection Measurement

A self-designed platform was used to study the single-wavelength diffuse reflection kinetics of samples under and after 375 nm laser irradiation, referring to relevant studies [33] for detailed description of the equipment. During the diffuse reflection kinetics measurement of methanol atmosphere, methanol-containing $N_2$ flowed continuously through the sample cell to capture photogenerated holes and facilitate the accumulation of photogenerated electrons. The temperature varies between 20 °C and 80 °C and remains constant during the reflection measurements. The reflection signal was first monitored in darkness for a period of time, and then measured continuously under 375 nm laser light. After the laser was turned off, the reflection is tested in the dark for 5 min. Finally, a high purity $O_2$ atmosphere was passed through the cell at a flow rate of 4 NL/min and the reflections were further monitored. The diffusion reflection kinetics measurement procedure in an $N_2$ atmosphere is the same as above, except that the pure $N_2$ in the atmosphere does not contain methanol. The measured diffusion reflectances were converted to the time-varying light absorption $A(t)$ of the sample according to the following formula:

$$A(t) = 1 - \frac{R(t)}{R_{ref}}' \tag{3}$$

where $R(t)$ is the diffusion reflectance of the sample over time, and $R_{ref}$ is the diffusion reflection of $BaSO_4$ under the same conditions.

### 3.5. Photocatalytic Experiments

The gas-phase photocatalytic performances of all samples were assessed by studying the acetone oxidation and isopropyl alcohol (IPA), both common air pollutants. The photocatalytic experiments for acetone oxidations were carried out in a self-designed closed-loop cylindrical batch reactor of pure quartz glass, referring to our previous study [34]. The photocatalytic reactor was heated to a set temperature using a heating plate, varying between 20 and 60 °C with an interval of 20 °C. A mercury lamp with a 365 nm band-pass filter was used as a light source. The light at 365 nm ensures that only $TiO_2$ was excited, so the role of Ag can be discussed. The light intensity was determined by the above silicon diode photodetecto connected to the power meter and was maintained at ~ 3 mW/cm$^2$ for all experiments.

The release of $CO_2$ from L-Ag/$TiO_2$, S-Ag/$TiO_2$, and $TiO_2$ samples during acetone oxidation was used to evaluate the photocatalytic activity. The release of $CO_2$ follows zero-order kinetics, so the formation rate was used to evaluate the photocatalytic activity. Prior to the photocatalytic reaction, the samples were first treated with ultraviolet light irradiation for a period of time to remove surface carbonate contaminants. Clean air then flowed through the reactor for ~15 min until the residual $CO_2$ concentration was below 20 ppm. The sample was first kept in dark at a set temperature for 30 min and then illuminated with a light for 30 min. One hour after the initial measurement, the UV lamp was turned off immediately, and then

5 μL acetone was injected into the reactor rapidly. After 30 min, the UV lamp was turned on and the test was continued for 30 min before the end of the measurement. Acetone and $CO_2$ concentrations were analyzed using an acoustic-photoelectric infrared multi-gas monitor (LumaSense Technologies INNOVA 1312).

For the UV photocatalytic IPA experiments, 50 mg sample powder was dispersed in 2 mL ethanol solution. After ultrasonic dispersion, a uniform slurry was formed and then dispersed on a quartz glass, which was used as the photocatalyst. The sample was dried in $N_2$ flowing at 50 °C. The catalytic experiments were carried out in a self-designed quartz glass reactor placed on a heating plate. A 300 W Xe lamp equipped with a 365 nm optical band pass filter was used as the light source, and the light intensity was also checked using the above silica-based photodetector. Under the control of mass flow meters, pure $N_2$ and IPA-$O_2$ standard gases first flowed through the reactor for about 30 min with a flow ratio of 5:1. Finally, the reactor was connected to a gas chromatograph (Shimadzu 2016) to form a sealed reaction measurement system, and then a series of IPA dehydrogenation experiments were performed at intervals of 10 °C between 30 and 50 °C. In the experiment, the reaction gases' concentrations were checked every 9 min. The conversion rates of IPA were used to evaluate the catalytic activity.

## 4. Conclusions

The electron relaxation kinetics of the undecorated $TiO_2$ and Ag/$TiO_2$ were studied under UV light illumination. Independent of the presence of methanol, the photocurrent and photo absorptions results showed that the electron relaxations mainly occur through the transfer to $O_2$ for the undecorated $TiO_2$. For the Ag/$TiO_2$, it was seen that the electron relaxation still occurs through the transfer to the residual $O_2$ in pure $N_2$, and the $E_{app}$ of the electron transfer is greatly decreased as compared to the undecorated sample. When the methanol was added, it was interesting to see that the electron relaxation became negatively dependent on temperatures for the Ag/$TiO_2$, indicating that the electron relaxation mechanism should change and does not correspond to the electron transfer to the $O_2$; this is independent on the Ag loaded amount and size. We proposed that the methanol molecule over the Ag/$TiO_2$ perimeters might play the role of the bridge for the recombination of the holes trapped in the Ag/$TiO_2$ perimeters and the electron stored in the Ag nanoparticles. Opposite to our expectation, it was seen that the photocatalytic acetone and isopropanol oxidations were decreased after the Ag decoration. Therefore, the Ag-assisted rapid electron relaxation did not mainly contribute to the photocatalytic reactions but promoted the recombination around the Ag/$TiO_2$ perimeters through the $O_2$ adsorption–desorption-assisted recombination. The S-Ag/$TiO_2$ showed much faster relaxation rates due to the high dispersion of small Ag nanoparticles over the TiO2 surface and the good connection between them.

**Supplementary Materials:** The following supporting information can be downloaded at: https://www.mdpi.com/article/10.3390/catal13060970/s1, Figure S1: Schematic diagram of the in situ conductance test platform; Figure S2: XRD patterns of $TiO_2$, L-Ag/$TiO_2$ and L-Ag/$TiO_2$ samples containing about 20 wt% Ag; Figure S3: U-I characteristic curves of pure $TiO_2$, L-Ag/$TiO_2$ and S-Ag/$TiO_2$ samples under dark conditions; Figure S4: The photocurrent curves of pure $TiO_2$, L-Ag/$TiO_2$ and S-Ag/$TiO_2$ in $N_2$ atmosphere with time before and after 375 nm laser irradiation; Figure S5: The photocurrent curves of pure $TiO_2$, L-Ag/$TiO_2$ and S-Ag/$TiO_2$ in methanol atmosphere with time before and after 375 nm laser irradiation; Figure S6: Absorption curves with time of pure $TiO_2$ and L-Ag/$TiO_2$ in $N_2$ atmosphere at 1550 nm before and after 375 nm laser irradiation; Figure S7: Absorption curves with time of pure $TiO_2$, L-Ag/$TiO_2$ and S-Ag/$TiO_2$ in methanol atmosphere at 1550 nm before and after 375 nm laser irradiation; Figure S8: The photocurrent curves of pure $TiO_2$, L-Ag/$TiO_2$ and S-Ag/$TiO_2$ in vacuum atmosphere with time before and after 375 nm laser irradiation; Figure S9: Curves of pure $TiO_2$, L-Ag/$TiO_2$ and S-Ag/$TiO_2$ at different temperatures for $CO_2$ release under UV irradiation.

**Author Contributions:** Experimental, Methodology, Data curation, Writing—original draft, W.Z.; Investigation and Validation, L.W.; Conceptualization, Supervision, project administration, funding acquisition, writing—review & editing, B.L. All authors have read and agreed to the published version of the manuscript.

**Funding:** B. Liu. thanks the National Key Research and Development of China (no. 2017YFE0192600) and the National Natural Science Foundation of China (no. 51772230). This work was also supported by the Guidance Project of Hubei Provincial Department of Education for Scientific Research (B2020246), the Research Fund for the Doctoral Program of Wuhan Technology and Business University (D2019008), and the Special Fund of Advantageous and Characteristic Disciplines (Group) of Hubei Province.

**Data Availability Statement:** The data presented in this study are available on request from the corresponding author.

**Conflicts of Interest:** The authors declare no conflict of interest.

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
