# Peer review of "Atmosphere-Dependent Electron Relaxation of the Ag-Decorated TiO2 and the Relations with Photocatalytic Properties"

_catalysts, doi:10.3390/catal13060970_

Round 1
Reviewer 1 Report
I've found this paper really interesting and well written. The information given in this article are original and of scientific impact. This is probably the first time that I recommend acceptance without substantial modification (only few grammar errors to be corrected).
I'm not a mother toungue, but it seems to me that the English quality is appropriate.
Author Response
Response: Thanks for highly evaluation on our work. The manuscript was polished and the grammar errors were corrected.
Reviewer 2 Report
The manuscript has some areas for improvement in presentation despite being generally intriguing. Here are some suggestions to enhance it:
1. To confirm the Ag decorated TiO2, the authors need to include in Figure 1 details of the TiO2 diffraction planes and their corresponding d-spacing value. In addition, lattice fringes are necessary to confirm the presence of Ag and TiO2.
2. Figure 2: The author should add the unit of absorbance to the figure.
3. Figure S1: The unit of intensity should be included in the figure.
4. The Introduction should clearly mention the significance of this paper.
5. Please explain why when Ag was added to TiO2, the XRD intensity was decreased.
6. If it’s possible to investigate the functional group of the final composite through FTIR analysis which can improve the quality of the manuscript.
7. Some editing for the English language is required throughout the manuscript due to too many mistakes
Before being accepted for publication, the submitted manuscript needs a minor revision
Author Response
Please see the attached file "response to reviewer2.pdf"

Reviewer 3 Report
Upon reviewing your manuscript intitled: Atmosphere-dependent electron relaxation of the Ag-TiO2 and the relations with photocatalytic properties. I find your work interesting, but I do not believe it can be published in its current form. Therefore, some revisions must be done so that it may be published in this journal. I have the following comments and recommendations.
-The introduction of relevant background and research progress was not comprehensive enough. Several recent works on TiO2 were not considered in the introduction.
- Doping plays a key role for any material's applications. Specifically, in TiO2 structures, several studies have shown the effectiveness of doping on photocatalytic properties. In relation to this, the authors do not justify the concentrations of dopants used. How is Ag entered distributed in the TiO2 structure? Does it enter the interstices or replace the Ti in the host lattice?
- On Figure 1, there are several diffraction peaks that were not identified. Should crystallographic card for the identified planes be included?
- How does Ag modify the crystalline structure of TiO2, considering that it has a different oxidation state and ionic radius than Ti?
- In general, the work is good, but lacks quantified discussions. In its present form it is descriptive and some hypothetical comments since few references are used to justify the ideas. This should be corrected throughout the entire manuscript.
Author Response
Comment 1: The introduction of relevant background and research progress was not comprehensive enough. Several recent works on TiO2 were not considered in the introduction.
Response:We added the discussion of some recent works in the revised manuscript, please check P3 lines 12-21, and P4, lines 1-5.
Comment 2: Doping plays a key role for any material's applications. Specifically, in TiO2 structures, several studies have shown the effectiveness of doping on photocatalytic properties. In relation to this, the authors do not justify the concentrations of dopants used. How is Ag entered distributed in the TiO2 structure? Does it enter the interstices or replace the Ti in the host lattice?
Response:In current research, we mainly investigated the Ag nanoparticle decoration over TiO2 surfaces. The effect of doping was not considered. In addition, because deposition of Ag nanoparticles was facile and conducted at low temperature, we considered that the Ag does not enter TiO2 lattice as dopants. In addition, it is also possible that some Ag exist over TiO2 surface small clusters and single atom, but the amounts is much lower as compared to the Ag nanoparticles.
Comment 3: On Figure s1, there are several diffraction peaks that were not identified. Should crystallographic card for the identified planes be included?
Response:The XRD diffractions peaks were indexed in the revised manuscript, please check the revised Figure S1.
Comment 4: How does Ag modify the crystalline structure of TiO2, considering that it has a different oxidation state and ionic radius than Ti?
Response: As the Ag nanoparticles were deposited over TiO2 surfaces under facile conditions, we think that does not affect the crystalline structure of the TiO2, please the experimental section for material fabrications
- In general, the work is good, but lacks quantified discussions. In its present form it is descriptive and some hypothetical comments since few references are used to justify the ideas. This should be corrected throughout the entire manuscript
Response: In the revised manuscript, we had already gave some discussions on our research result, a Figure (Figure 9) that illustrate the photocatalytic mechanism was also added. In addition, we indeed found that the loading of Ag nanoparticles can increase the electron relaxation to O2, which is also same to the Au and Cu modified TiO2, especially in the presence of methanol. To the best of knowledge, few works devoted to investigating this issue. In the current research, we clearly gave detailed experimental proof and some discussion that Ag decoration can increase electron relaxation and change the mechanism. As the finding is the same for the Au and Cu modified TiO2, so we will gave a detailed investigation in our future work
Reviewer 4 Report
Brief Summary: The article presents the electron relaxation studies on Ag/TiO2 at different atmospheric conditions and evaluates the possible effect of Ag addition on the photocatalytic performance of TiO2 in light of the deduced mechanisms from the relaxation studies.
Strengths/Weaknesses of the manuscript/work: The study provides a logical methodology in assessing the (mechanistic) hypothesis, which is more apparent towards the end of the article. The title accurately captures the main text. However, the essence of this work could be made more obvious at the start of the article so that the readers can better appreciate its significance.
Specific comments:
1. Needs further English editing.
2. References should be cited here (lines 36-37): “For TiO2 materials, it had been found that the hole transfer to organics occurs on the timescale of ps to ns, while the electron transfer to O2 occurs on the timescale of ms to μs.”
3. Lines 45-48: But also, in the work of Dai, et al., they have a slower decay, in addition to the faster one. I think both should be mentioned since your work deals with photocatalytic reactions, where the latter (trapped hole capture by some species on the surface) seems more important.
4. Materials and Methods should be placed ideally before Results and Discussion.
5. Lines 79-102: In Fig. 2, it also seems to have a peak shift between S- and L-Ag/TiO2? I suggest to also indicate the peak positions and further interpret results here if possible. Can you deduce something from the difference in the peaks between L- and S-Ag/TiO2? And whether the observed properties make sense?
6. It is not clear from the introduction (and abstract why you have S- and L-Ag/TiO2).
7. In Figure S2, what are the dashed and solid lines (which one is the dark)? This should be pointed out. Also, the S-Ag/TiO2 shows a slight non-ohmic character?
8. Line 204: Should this really be Equation 3? I suggest to place the equation near the text it was mentioned.
9. Section 2.4: It would be better to show schemes/diagrams showing the explained mechanism in the text.
§ Lines 279-282: Maybe one should also consider the Temperature-dependence of methanol adsorption (on Ag/TiO2).
10. Lines 285-286: One has to cite the paper (or is this unpublished)? (your own Au/TiO2 result)
11. The conclusion is clear and justified.
12. The references can be updated to more recent ones.
The English needs further editing to make it easier to read. The text is generally comprehensible but some errors in grammar can slow down the reader.
Author Response
Comment 1: Strengths/Weaknesses of the manuscript/work: The study provides a logical methodology in assessing the (mechanistic) hypothesis, which is more apparent towards the end of the article. The title accurately captures the main text. However, the essence of this work could be made more obvious at the start of the article so that the readers can better appreciate its significance.
Response:We illustrated the significance in the revised manuscript, please check P5, lines, and P6 lines.
Comment 2: Needs further English editing.
Response:Grammar accuracy and sentence fluency have been corrected.
Comment 3: References should be cited here (lines 36-37): “For TiO2 materials, it had been found that the hole transfer to organics occurs on the timescale of ps to ns, while the electron transfer to O2 occurs on the timescale of ms to μs.”
Response:The relevant paper has been cited.
Comment 4: Lines 45-48: But also, in the work of Dai, et al., they have a slower decay, in addition to the faster one. I think both should be mentioned since your work deals with photocatalytic reactions, where the latter (trapped hole capture by some species on the surface) seems more important.
Response:The relevant discussion was added in the revised manuscript, please check P4 lines 11-15
Comment 5: Materials and Methods should be placed ideally before Results and Discussion.
Response:The arrangement of the article has been adjusted according to the requirements.
Comment 6: Lines 79-102: In Fig. 2, it also seems to have a peak shift between S- and L-Ag/TiO2? I suggest to also indicate the peak positions and further interpret results here if possible. Can you deduce something from the difference in the peaks between L- and S-Ag/TiO2? And whether the observed properties make sense?
Response:The plasma resonance absorption peaks of L-Ag/TiO2 and S-Ag/TiO2 are 470 nm and 484 nm, respectively. However, it was seen that the plasmon absorption of S-Ag/TiO2 does not blue-shift as compared to the L-Ag/TiO2; this apparently is not inconsistence with the plasmon theory; this might be relatively easy oxidation of the small-sized Ag nanoparticles. The difference in the plasmon absorption does not affect the results and the conclusions of the current research.
Comment 7: It is not clear from the introduction (and abstract why you have S- and L-Ag/TiO2).
Response: The effect of the Ag nanoparticle size was discussed in the abstract and the revised manuscript. Please check. The principle of the Ag nanoparticle to increase the electron relaxation does not depend on the particle size. The higher relaxation rate for the S-Ag/TiO2 should be ascribed to the high dispersion over TiO2 surface and the large connection with TiO2 nanoparticles.
Comment 8: In Figure S2, what are the dashed and solid lines (which one is the dark)? This should be pointed out. Also, the S-Ag/TiO2 shows a slight non-ohmic character?
Response:The solid line represents the curve of the forward voltage scanning, and the dashed line represents the curve of the reverse scanning voltage. Actually, we do not know the detailed reason for slight deviation from the ohmic feature, might be the sample preparation. However, on the whole, the IV curve can follow the linear dependence, based on which, our conductance analysis should be believable.
Comment 9: Line 204: Should this really be Equation 3? I suggest to place the equation near the text it was mentioned.
Response:It has been corrected in the corresponding position in the article.
Comment 10: Section 2.4: It would be better to show schemes/diagrams showing the explained mechanism in the text.
Response:The relevant mechanism diagram has been drawn in section 3.5 of the revised article. Please check the Figure 9. In P 27.
Comment 11: Lines 279-282: Maybe one should also consider the Temperature-dependence of methanol adsorption (on Ag/TiO2).
Response: Yes, we considered the temperature should have an effect on the methanol absorption on TiO2 surface. It is also possible that the temperature variation affects absorption capacity over the Ag/TiO2 interface and finally reduce the electron relaxation rate. This is good comment. As we also observed such results for Au/TiO2 and Cu/TiO2, we will give a detail investigation on this topic in a future work. As we do not clearly know the detailed physical reason for this, so we did add this in the revised manuscript.
Comment 12: Lines 285-286: One has to cite the paper (or is this unpublished)? (your own Au/TiO2 result)
Response:Relevant paper has been cited in the corresponding places in the article. Please check P23. Lines 10.
comment 13: The references can be updated to more recent ones.
Response: We update the recent literatures in the revised manuscript, please check P30, lines 25-34
Round 2
Reviewer 3 Report
This version can be accepted for publication